# Emulsifiers Impact Colonic Length in Mice and Emulsifier Restriction is Feasible in People with Crohn’s Disease

**DOI:** 10.3390/nu12092827

**Published:** 2020-09-15

**Authors:** Alicia M. Sandall, Selina R. Cox, James O. Lindsay, Andrew T. Gewirtz, Benoit Chassaing, Megan Rossi, Kevin Whelan

**Affiliations:** 1Department of Nutritional Sciences, School of Life Course Sciences, Faculty of Life Sciences and Medicine, King’s College London, London SE1 9NH, UK; alicia.sandall@kcl.ac.uk (A.M.S.); selina.cox@kcl.ac.uk (S.R.C.); megan.rossi@kcl.ac.uk (M.R.); 2Department of Gastroenterology, Barts Health NHS Trust, Royal London Hospital, London E1 1BB, UK; james.lindsay8@nhs.net; 3Centre for Immunobiology, Blizard Institute, Queen Mary University of London, London E1 2AT, UK; 4Centre for Inflammation, Immunity and Infection, Institute for Biomedical Sciences, Georgia State University, Atlanta, GA 30303, USA; agewirtz@gsu.edu (A.T.G.); benoit.chassaing@inserm.fr (B.C.); 5Neuroscience Institute, Georgia State University, Atlanta, GA 30303, USA; 6INSERM, U1016, Team “Mucosal microbiota in chronic inflammatory diseases”, 75006 Paris, France; 7Université de Paris, 75006 Paris, France

**Keywords:** food additives, emulsifiers, ultra-processed foods, feasibility study, inflammatory bowel disease, Crohn’s disease

## Abstract

There is an association between food additive emulsifiers and the prevalence of Crohn’s disease. This study aimed to investigate: (i) the effect of different classes of emulsifiers on markers of intestinal inflammation in mice and (ii) the feasibility, nutritional adequacy and symptom impact of restricting all emulsifier classes in Crohn’s disease. Mice were exposed to different classes of emulsifiers (carboxymethycellose, polysorbate-80, soy lecithin, gum arabic) in drinking water for 12-weeks, after which markers of inflammation and metabolism were measured. A low emulsifier diet was developed to restrict all classes of emulsifiers and its feasibility measured over 14-days in 20 participants with stable Crohn’s disease. Crohn’s disease-related symptoms, disease control, body weight and composition, nutrient intake and food-related quality of life (QoL) were measured. All emulsifiers resulted in lower murine colonic length compared with control (mean 9.5 cm (SEM 0.20)), but this only reached significance for polysorbate-80 (8.2 cm (0.34), *p* = 0.024) and carboxymethylcellulose (8.0 cm (0.35), *p* = 0.013). All 20 participants completed the feasibility study. The frequency of consuming emulsifier-containing foods decreased by 94.6% (SD 10.3%). Food-related QoL improved between habitual (median 81.5 (IQR 25.0)) and low emulsifier diet (90.0 (24.0), *p* = 0.028). Crohn’s disease-related symptoms reduced (median 3.0 (IQR 5.3) vs. 1.4 (3.9), *p* = 0.006), and disease control scores improved (13.5 (IQR 6.0) vs. 15.5 (IQR 3.0), *p* = 0.026). A range of emulsifiers may influence intestinal inflammation in mice, and dietary restriction of emulsifiers is feasible. Trials investigating the efficacy of a low emulsifier diet in Crohn’s disease are warranted.

## 1. Introduction

Crohn’s disease is increasing in prevalence globally, and this within-generation increase implicates lifestyle factors in disease aetiology [1]. Economic development and industrialisation are associated with adoption of a ‘‘Westernised’’ lifestyle characterised by diets high in processed foods, which have been correlated with increased Crohn’s disease incidence [1,2,3,4]. Furthermore, the reintroduction of normal diet following enteral nutrition-induced remission of Crohn’s disease [5,6] is associated with disappointingly high relapse rates [7], incriminating dietary components in the natural history of Crohn’s disease [8]. In addition, 70% of patients report that diet affects disease activity [9]. There is inconsistent evidence that macronutrients such as sugar [10,11,12] and fat [5,11,12] impact intestinal inflammation; however, food-additive emulsifiers, a ubiquitous component of the ”Westernised” diet, have been implicated in intestinal inflammation and Crohn’s disease via their impact on intestinal microbiota, barrier function and immunity [13].

Emulsifiers are a type of food additive used to provide a stable and uniform consistency in fat-containing foods that would otherwise separate into oil and water. Emulsifiers have many applications in processed food items including retaining water for freshness, strengthening dough crumb texture, reducing melting rates of frozen foods, lengthening shelf life of cream-based foods and behaving as vegetarian gelling agents as an alternative to animal gelatine [14,15]. Emulsifiers therefore optimise the appearance, texture and mouthfeel of foods and have become a ubiquitous part of the diet in economically developed countries [4]. Emulsifiers can be categorised based on their chemical structure and subsequent effects in foods into: (1) low molecular weight surfactants (e.g., polysorbate-80 or lecithin) with opposing charges within the molecule that inhibit aggregation of fat droplets; (2) amphiphilic biopolymers (e.g., gum Arabic) that adsorb to the oil-water interface and (3) thickeners/stabilisers (e.g., carboxymethylcellulose) that increase viscosity, creating a gel and thus stabilising emulsions [16,17].

Evidence implicating emulsifiers in the aetiology of inflammatory bowel disease (IBD) is accumulating. One study used human intestinal epithelial cell lines, including HCT-8 (ileocecum) and Caco-2 (colonic), and showed that polysorbate-80 and carrageenan increase bacterial translocation through disruption of epithelial tight junctions [18,19,20,21,22,23]. Carrageenan also increased transcription factors, such as NF-κB, that signal production of inflammatory cytokines including IL-8 [22]. A landmark murine study reported that polysorbate-80 and carboxymethylcellulose altered mucus layer homeostasis and increased intestinal inflammation in wildtype, IL-10^-/-^ and TLR-5^-/-^ knockout mice, although only induced colitis in knockout mice [13]. This supports evidence that IBD aetiology involves both genetic susceptibility and an environmental trigger, such as emulsifier exposure. In a follow-up study using an in vitro gut simulator model with human microbiota, polysorbate-80 and carboxymethylcellulose were found to increase the inflammatory potential of the microbiota (by increasing bioactive flagellin). Altered microbiota composition in these mice was characterised by an increase in bacteria putatively associated with inflammation, such as Proteobacteria and Enterobacteriaceae, and decreased Bacteroidaceae [24], and such altered microbiota were sufficient to induce intestinal inflammation when transplanted to germfree recipient mice.

These in vitro and in vivo studies implicate three emulsifiers, polysorbate-80 (a low molecular weight surfactant), carrageenan and carboxymethylcellulose (thickeners and stabilisers), in gastrointestinal inflammation. However, there are 65 emulsifiers present in the global food supply. This presents two key challenges in research and diet in IBD [25]. Firstly, greater understanding of the breadth of emulsifier-induced inflammation is required, given that other emulsifiers, including those from different classes used more commonly, such as soy lecithin (low molecular weight surfactant) and gum arabic (amphiphilic biopolymer), have yet to be tested for their proinflammatory potential. Secondly, if a range of emulsifiers cause gastrointestinal inflammation, it would be important to assess whether a low emulsifier diet is feasible and effective as a therapeutic intervention in Crohn’s disease.

Daily exposure to dietary emulsifiers is high [26], but emulsifier presence in foods may not be obvious to the consumer and reading food labels to correctly identify hundreds of variations of emulsifier names and E-numbers would be challenging. A diet low in emulsifiers is a novel intervention, and the degree of dietary behaviour change required to strictly adhere to the diet, such as changing food shopping, meal planning, food preparation and eating out, is unknown. Whilst a previous study has advised on a low carrageenan diet in IBD [27], this diet removed only one emulsifier; thus, participants could be taught label-reading to exclude this one ingredient. Furthermore, imposing a low emulsifier dietary restriction on a patient group that already limits certain food groups may exacerbate nutrient inadequacy [28].

This study aims to investigate (i) the effect of different classes of emulsifiers on markers of intestinal inflammation in mice and (ii) the feasibility, acceptability and nutritional adequacy of restricting all classes of emulsifiers in people with Crohn’s disease. This study is the first to explore the potential of a range of classes of emulsifiers and the feasibility of a low emulsifier diet, opening research regarding the potential of this diet as a low cost and safe therapy that could change the way Crohn’s disease is managed.

## 2. Materials and Methods

### 2.1. Murine Study

A 12-week controlled emulsifier feeding experiment was performed on 25 adult male, wild-type C57BL/6 mice (Jackson Laboratories, Bar Harbor, ME, USA) bred and housed at Georgia State University (Atlanta, GA, USA). Mice were 4–5 weeks old and initial bodyweight was mean 12 g.

All mice were fed Purina rodent chow (#5001) and water ad libitum. The same drinking water (reverse-osmosis treated Atlanta city water) was used for all groups. Five groups of mice (*n* = 5) were exposed to either drinking water with no additions (control) or one of four emulsifiers from varying classes at a concentration of 1.0% for 12 weeks. The emulsifiers were polysorbate-80 (low molecular weight surfactant), soy lecithin (low molecular weight surfactant), gum arabic (amphiphilic biopolymer) and sodium carboxymethylcellulose (thickener/stabiliser) (average M_W_ ~250,000) and were chosen to represent a range of different types of emulsifier present in the human food supply, encompassing two that have previously been investigated (sodium carboxymethylcellulose and polysorbate-80) [13] and two widely used emulsifiers with as yet unknown impacts on inflammation (soy lecithin and gum arabic). Emulsifiers were sourced from Sigma-Aldrich (St Louis, MO, USA).

Body weight was measured weekly. After 12 weeks of emulsifier or control treatment, 24-h food intake was assessed. Groups of mice were placed in a clean cage with a known amount of food. Twenty-four hours later, amount of remaining food was measured to calculate 24-h food intake per mouse. For overnight fasting blood glucose measurement, mice were fasted for 15 h, and blood glucose concentration was determined using a Nova Max Plus Glucose Meter. Finally, mice were fasted for 5 h, euthanized, and colonic length and weight, spleen weight, adipose weight and caecum weight were measured to deduce digestive organ changes [13].

### 2.2. Human Feasibility Study

#### 2.2.1. Study Design and Participants

This was a feasibility study in patients with a diagnosis of Crohn’s disease conducted between July and December 2018 (ClinicalTrials.gov Identifier: NCT03691155). Feasibility, adherence, acceptability, nutritional adequacy and symptom impact of a low emulsifier diet were measured.

Participants were recruited online through the charity *forCrohns* (UK Registered Charity Number 1129871). Adults aged ≥18 years with a diagnosis of stable Crohn’s disease (no change in medication in the last 3 months, no surgery or hospital admission related to Crohn’s disease in the last 6 months and no new perianal Crohn’s disease in the last 6 months) were recruited. All participants had to own an Android or IOS (Apple) smartphone in order to use a smartphone application to follow the diet. Exclusion criteria were current treatment with corticosteroids or exclusive or partial enteral nutrition; prescription of oral nutrition supplements for nutrition support; unexplained/unintentional weight loss in the past 6 months and BMI < 18.5 kg/m^2^, since participation would require dietary restrictions that could exacerbate nutritional inadequacy.

Baseline demographic data and clinical data on duration since diagnosis, age, sex, smoking status, Crohn’s disease medications and alternative and complementary therapy use were collected.

As this was a feasibility study, a formal sample size calculation was not appropriate [29]. A sample size of 20 participants was justified a priori based on the study objectives. This sample size was based on published recommendations of a minimum sample size of 12 participants for feasibility studies and agreed in consensus with the research steering committee (IBD clinicians and researchers) as sufficient to highlight major feasibility issues of following the low emulsifier diet [30].

#### 2.2.2. Intervention

The intervention was a 14-day low emulsifier diet designed to exclude 65 food additives classified as emulsifiers based on both JECFA (Joint FAO/WHO Expert Committee on Food Additives) and Codex Alimentarius [31,32] (see Appendix A). The diet excluded all food additive emulsifiers including those in the murine feeding study: polysorbate-80 (E433), soy lecithin (E322), gum arabic (E414) and carboxymethylcellulose (E466).

The low emulsifier diet was delivered through dietetic counselling, an educational booklet and a novel smartphone application (app). During the counselling session with the same registered dietitian (AS), participants’ habitual diet was reviewed, in line with routine dietetic practice, so that suitable alternative emulsifier-free foods could be suggested. A diet information booklet was provided including suitable/unsuitable food lists, recipes and shopping advice, and participants were given access to the app (hosted by FoodMaestro, UK Registered Company No: 4093976, currently not commercially available and for research purposes only). The app included a barcode scanner function that informed the participant if a food was suitable (emulsifier-free) or unsuitable (emulsifier-containing), and participants were instructed to scan all foods purchased or consumed with a barcode. If the barcode could not be read by the app, participants were advised to avoid the food. Use of the app meant suitable foods could be easily identified without participants needing to label-read hundreds of variations of emulsifier names and E-numbers. Therefore, participants were only told that the diet excluded types of ingredients, rather than emulsifiers specifically. Blinding participants in this way meant that the feasibility of a future blinded randomised control trial (RCT) could be assessed.

Participants were instructed to follow the low emulsifier diet for 14 days. The dietitian contacted participants three to four days later to provide support, encourage dietary adherence and address concerns or issues raised following the first few days of shopping, cooking and consuming a low emulsifier diet. Participants were able to contact the study dietitian with any questions throughout the study period. Participants returned for their end of intervention visit and feasibility outcomes were recorded.

#### 2.2.3. Outcome Measurement

Dietary intake was measured using a using 7-day food diary at baseline (7-days before commencing the diet) and at the end of intervention (final 7-days of the 14-day intervention period). Patients recorded all food and drink intake including brand name, quantity consumed (using a food portion photo guide), cooking method and time of eating episode, in line with gold standard dietary assessment [33].

Adherence to the low emulsifier diet was measured as emulsifier intake recorded in the 7-day food diary. Quantifying absolute intakes of emulsifiers (for example in mg/day) was not possible as there are currently no food composition databases that list emulsifier content in foods. Furthermore, food label regulations require only that the presence, and not quantity, of emulsifiers be listed [34]. Therefore, 12 major food companies were contacted for indicative product emulsifier content of food products but were unwilling to release this commercially sensitive data. Therefore, emulsifier intake was recorded as the frequency of consumption of food items containing emulsifiers. To assess emulsifier exposure, the label of every food/drink reported in the 7-day food diary was checked to confirm the presence/absence of emulsifiers. A food label confirming emulsifier content was defined as “definite exposure”, but where no food label was available (e.g., takeaway meals), the food was assessed by the dietitian and defined as “probable exposure” where relevant (for example, many Chinese takeaway dishes contain emulsifiers due to processed sauces). Adherence was defined as a 75% reduction in frequency of emulsifier intake from baseline to end of intervention. In order to measure the distribution of exposure, the number of emulsifier-containing eating episodes was compared between baseline and end of intervention. The definition of an “eating episode” was consumption of ≥50 kcal at least 30 min apart from another eating occasion, in line with standard definitions [35,36]. Adherence was also recorded subjectively by participants using a Likert scale at the end of intervention visit (reporting to have followed the diet 0%, 25%, 50%, 75% or 100% of the time).

The impact of the low emulsifier diet on nutrient intake was also assessed by coding the 7-day food diaries using a nutritional analysis software (Nutritics^®^ Research Edition v5.02, Dublin, Ireland), which incorporates data from the UK Composition of Foods [37]. The proportion of participants meeting national dietary targets for each nutrient [38,39,40,41,42] was compared before and during the low emulsifier diet.

The impact of the low emulsifier diet on anthropometry was assessed at baseline and end of intervention. Height was measured using a wall mounted stadiometer with head position in the Frankfurt plane [43] and weight measured in light indoor clothes without shoes and after voidance of urine [43], after which body mass index (BMI) was calculated (kg/m^2^). Bioelectrical impedance analysis was used to assess the impact of the diet on fat mass and lean body mass (Tanita BC-418MA).

Feasibility and acceptability of the low emulsifier diet (including the delivery of the diet, the app, the booklet) were measured at the end of the intervention using a questionnaire based on an adapted Education Method Usability Scale [44] (“strongly agree” to “strongly disagree” Likert scale) and open questions (analysed thematically).

The impact of the low emulsifier diet on the psychosocial aspects of food, nutrition, eating and drinking was assessed using the FR-QoL-29at baseline and end of the intervention. This food-related quality of life (QoL) questionnaire was developed and validated in IBD and comprises 29 statements with Likert scale responses [45,46].

The impact of the diet on Crohn’s disease-related symptoms was recorded at baseline and end of intervention using two questionnaires. The Patient-Reported Outcome-2 (PRO-2) questionnaire has been validated for use in Crohn’s disease, correlates with CDAI and may be used as a proxy for disease activity [47]. The higher the PRO-2 score, the greater the symptom severity, with scores <8 associated with disease remission [47]. The IBD Control-8 questionnaire (IBD-C-8) was also used at baseline and end of intervention to measure overall patient-perceived disease control and to confirm no detrimental impact of the diet on gastrointestinal symptoms [48]. This questionnaire contains eight multiple-choice items, with a score ≥13 (out of 16) being consistently associated with disease remission [48].

### 2.3. Statistical Analysis

Data from the murine feeding study are presented as mean and standard error of the mean (SEM). Significance was determined using one-way ANOVA with Bonferroni’s post hoc correction for multiple comparisons between groups. Principal coordinate analysis was performed to integrate the effect of individual emulsifiers on inflammatory and metabolic outcomes. Analysis was performed using GraphPad Prism (version 6.01, GraphPad Software, La Jolla, California City, CA, USA).

Continuous data from the human feasibility study are presented as mean and standard deviation (SD) or as median and interquartile range (IQR), according to data distribution and were compared between baseline and end of intervention using a paired *t*-test (normally distributed data) or Wilcoxon-signed rank (non-normally distributed data). Categorical data are presented as frequencies and percentages and were compared between baseline and end of intervention using McNemar’s test (dichotomous categories) or Wilcoxon-signed rank test (three or more categories). Emulsifier exposure co-occurrence in the diets of participants was investigated using Spearman’s rank correlation and presented as a heatmap. Statistical analyses were performed using SPSS^®^ for Windows^®^ version 22 (IBM Corp., Armonk, NY, USA). For all statistical tests, a *p*-value of <0.05 was considered statistically significant.

### 2.4. Ethical Considerations

Mice were bred and housed at Georgia State University (Atlanta, GA, USA) under institutionally approved protocols (Institutional Animal Care and Use Committee #A14033 and #A18006). The human feasibility study was approved by the Research Ethics Committee of King’s College London (Ethical Clearance Reference Number: LRS-17/18-7313). Written informed consent was obtained from all participants. The feasibility study was registered on ClinicalTrials.gov registry on 8th July 2018 (NCT03691155).

## 3. Results

### 3.1. Murine Study

There was no difference in caecal weight in the emulsifier-treated mice compared to control (Figure 1a), while colonic weight was reduced in all emulsifier-treatment groups compared to control, with this only reaching statistical significance in the polysorbate-80 treated mice (254.4 mg SEM 5.7 mg vs 318.3 mg SEM 7.6 mg, *p* < 0.001) (Figure 1b). Colon shortening, a proxy marker of intestinal inflammation in mice, was observed in all emulsifier-treated mice compared to control (Figure 1c), with statistically significant shorter colonic length found in carboxymethylcellulose (8.0 cm SEM 0.35, *p* = 0.013) and polysorbate-80 (8.2 cm SEM 0.34, *p* = 0.024) treated mice compared with control (9.5 cm SEM 0.20) (Figure 1c).

Consumption of emulsifiers carboxymethylcellulose and polysorbate-80 led to an increase in body weight after 12 weeks of exposure, mainly driven by an increase in fat deposition (Figure 2a,b). Food intake was greater in carboxymethylcellulose and polysorbate-80 treated mice compared with control; however, this only reached statistical significance in the polysorbate-80 treated mice (4.23 g/day SEM 0.05 vs 5.61 g/day SEM 0.07, *p* = 0.009, Figure 2c). While a tendency to increased fasting blood glucose level was observed in emulsifier-treated mice, significance was not reached compared with water-treated mice (Figure 2d).

When combining these various measurements using principal coordinate approach, it appears that carboxymethylcellulose and polysorbate-80 have strong effects on inflammatory and metabolism-related parameters, while soy lecithin seems to have minimal impact compared to water-treated animals, and the effects of gum arabic appear highly variable and intermediate (Figure 3).

### 3.2. Feasibility Study

Twenty participants with Crohn’s disease were recruited to the feasibility study, and all completed the low emulsifier diet intervention (Figure 4). The mean age of participants was 34.9 years (SD 10.1), and the majority were female (14/20, 70%) (Table 1).

#### 3.2.1. Emulsifier Exposure in Habitual Diet in Crohn’s Disease

The 7-day food diaries were analysed at baseline for habitual emulsifier intake. All participants consumed emulsifiers in their habitual diet; the majority (15/20, 75%) consumed emulsifiers every day, with the remaining 5 participants (5/20, 25%) consuming emulsifiers on 6 out of 7 days in the baseline week. Of the 65 food-additive emulsifiers, 33 were consumed by participants in the baseline week. At baseline, the emulsifier-containing foods consumed by patients with Crohn’s disease contained a median (minimum–maximum) of 2 (1–10) emulsifiers per product.

The emulsifiers most frequently consumed were lecithin (1.31 /day, SD 1.11), mono- and diglycerides of fatty acids (0.99, SD 1.42), pectin (0.49, SD 0.62), diacetyl tartaric acid esters of mono- and diglycerides (0.47, SD 0.62) and xanthan gum (0.43, SD 0.52). In terms of the emulsifiers tested in the murine study, the frequency of exposure in patients with Crohn’s disease was polysorbate-80 0.01/day (SD 0.03), carboxymethylcellulose 0.14/day (SD 0.30) and gum arabic 0.09/day (SD 0.15). Daily exposure to carrageenan was 0.09/day (SD 0.15). There was a strong positive correlation between frequency of exposure to cross-linked sodium carboxymethycellulose and tragacanth gum (rho = 1.00, *p* < 0.01), between cross-linked sodium carboxymethycellulose and tartaric esters of mono- and di-glycerides of fatty acids (rho = 1.00, *p* < 0.01) and between tragacanth gum and tartaric acid esters of mono- and di-glycerides of fatty acids (rho = 1.00, *p* < 0.01) (Figure 5).

Of the 422 emulsifier-containing foods consumed by participants at baseline, the most common sources of emulsifiers were cereal and cereal products (177/422, 42%), sugar preserves and confectionary (83/422, 19.7%) and miscellaneous (53/422, 12.6%) (Table 2). The following food groups did not contribute to emulsifier intake at baseline: “eggs and egg dishes”, “fruit” or “alcoholic beverages”. The breakdown of food group contribution to emulsifier exposure is shown in Table 2.

#### 3.2.2. Adherence to the Low Emulsifier Diet

Good adherence to the low emulsifier diet was defined a priori as at least 75% reduction in the frequency of emulsifier intake between baseline and end of intervention, and this was achieved by 95% (19/20) of patients. The mean change in frequency of emulsifier intake during the trial was −94.6% (SD 10.3%), demonstrating overall high dietary adherence. The number of emulsifier-containing eating occasions significantly decreased from a median of 2.3 per day (IQR 1.5) at baseline to 0.0 per day (IQR 0.1) during the diet (*p* < 0.001).

Eighteen participants (90%) reported following the diet 100% of the time. The remaining two reported following the diet 75% of the time, and the reason stated for reduced adherence by both participants was needing to eat out. Eight participants (40%) reported eating a non-permitted food during the trial. An open question explored which unpermitted foods were consumed, and responses were themed into the following groups (descending order): restaurants/eating outside of the home; dairy and dairy-based desserts; biscuits, cakes and confection; and baked goods.

#### 3.2.3. Nutrient Intake, Anthropometry and Crohn’s Disease-Related Symptoms during Low Emulsifier Diet

During the low emulsifier diet there were statistically significant reductions in intakes of energy, carbohydrate, saturated fat, sodium, calcium, niacin and vitamin B_12_ compared with baseline habitual diet (Table 3). However, these changes were not clinically significant as the proportion of participants meeting national recommended requirements was not significantly different for any of these nutrients (Table 3). Calcium intakes were reduced by only 69 mg (from 813 mg (IQR 483) to 744 (IQR 178), *p* = 0.015); however, there were no differences in the number and proportion achieving national dietary recommendation of 700 mg/day (13 (65%) vs. 12 (60%), *p* = 1.000) or the higher recommended intake for IBD of 1000 mg/day (6 (30%) vs. 3 (15%), *p* = 0.375) [39,41]. Intake of alcohol was lower during the low emulsifier diet. There were no significant changes in intakes of any other nutrients during the low emulsifier diet, either in absolute terms or in the proportions meeting national recommendations, for any other nutrient (Table 3).

There were no significant changes between baseline and low emulsifier diet, respectively, in weight (67.7 kg (IQR 22.7) vs. 68.8 kg (IQR 22.2), *p* = 0.387) body mass index (BMI 23.2 kg/m^2^ (IQR 4.0) vs. 23.3 kg/m^2^ (IQR 4.2), *p* = 0.445) or percent fat free mass (73.2% (IQR 14.2) vs. 73.6% (IQR 11.7), *p* = 0.209).

Crohn’s disease-related symptoms, as assessed by the PRO-2, improved between baseline and low emulsifier diet (3.0 IQR 5.3 vs. 1.4 IQR 3.9, *p* = 0.006). The proportion of participants in disease remission based on PRO-2 did not change (17 (85%) vs. 18 (90%), *p* = 0.157). Additionally, patient-perceived disease control, as assessed by the IBD-C-8, increased (13.5 IQR 6.0 vs. 15.5 IQR 3.0, *p* = 0.026), which is clinically significant [48].

#### 3.2.4. Feasibility and Acceptability of the Low Emulsifier Diet

After dietary counselling on how to follow the low emulsifier diet, half of participants (10/20, 50%) did not need to seek further advice from the dietitian during the two weeks, with the remainder contacting the dietitian between one and three occasions with queries. The majority of participants (16/20, 80%) reported using the app once or more per day during the diet, with the remainder using the app 4–5 times per week (2/20, 10%), with the least frequent usage reported at 2–3 times per week (2/20, 10%). The majority of participants felt confident using the mobile app and felt others could learn to use the app quickly (19/20, 95%). The majority of participants (18/20, 90%) reported finding the diet more difficult to follow than their normal diet; however, most deemed the diet as appetising (19/20, 95%) (see Appendix A).

Suggestions to help future participants follow the diet were themed into the following areas: planning meals and snacks during the diet, ensuring accessibility to suitable food products, educating yourself on the suitable and unsuitable foods, taking time when shopping and buying in bulk and avoiding eating out and instead preparing own food. The foods considered most inconvenient to avoid were in relation to the theme of restaurants, eating out and meals prepared by friends and family (as these foods could not be scanned and so had to be avoided).

Food-related-QoL significantly improved on the low emulsifier diet from a baseline median score of 81.5 (IQR 25.0) to 90.0 (IQR 24.0) at end of intervention (*p* = 0.028).

## 4. Discussion

The safety testing of dietary emulsifiers has historically focused on toxic effects or serious adverse events in acute animal feeding studies [50]. However, for the majority of emulsifiers, the effects on intestinal inflammation have not been investigated in animals, and the effects on intestinal inflammation in patients with Crohn’s disease have not been examined for any emulsifiers. Moreover, the feasibility of implementing a therapeutic low emulsifier dietary intervention is unknown.

The murine study confirmed the inflammatory potential of various dietary emulsifiers. Reduction in colon length, a proxy measure of intestinal inflammation, was observed following treatment with all the tested emulsifiers, although only carboxymethylcellulose and polysorbate-80 reached statistical significance; the principal coordinate analysis showed gum arabic had highly variable effects. Previous murine studies have demonstrated the colitis-inducing effects of carboxymethylcellulose and polysorbate-80 [13,24], but this is the first time other commonly consumed emulsifiers, lecithin and gum arabic, have been tested for their gastrointestinal impact in a murine model. The feasibility study demonstrated that soy lecithin and gum arabic were more commonly consumed than carboxymethylcellulose and polysorbate-80 in the habitual diet of patients with Crohn’s disease. Given that emulsifiers other than carboxymethylcellulose and polysorbate-80 approached significance for reducing colon length in this study, and their consumption frequency was high, an investigation of the effectiveness of restricting all emulsifier classes in Crohn’s disease is warranted. A limitation of the murine study is that only a proxy marker of intestinal inflammation, colonic length, was measured.

The feasibility study demonstrated for the first time that a low emulsifier diet, when delivered by a dietitian, was acceptable and safe in people with Crohn’s disease. While other exclusion diets have been devised for the treatment of Crohn’s disease [51,52], this is the first to specifically target and exclude one group of food additives based on the convincing murine and in vitro studies [13,18,19,20,21,22,23,24]. The observed high rates of adherence and acceptability of the low emulsifier diet makes it a feasible therapeutic intervention for people with Crohn’s disease. Whilst exclusive enteral nutrition can induce remission in Crohn’s disease, poor adherence is a limiting factor in its success [5], and the unpalatability of exclusive enteral nutrition is a common reason for non-adherence and a common cause of cessation [53]. In contrast, 95% of participants in the present study were considered compliant with the diet, and the same number reported that the low emulsifier diet was as appetising as their normal diet.

Prior to this feasibility study, the degree of dietary change required for people with Crohn’s disease to strictly adhere to a low emulsifier diet was unknown. We report in this small sample of people with Crohn’s disease that participants were habitually exposed to multiple dietary emulsifiers daily, suggesting a diet eliminating these food additives would involve extensive food and lifestyle changes. Given that 65 food additives can be classified as emulsifiers, a further challenge is a patient’s ability to identify vast numbers of emulsifiers in foods, while understanding the widespread variation in their presence in food items (for example, some brands of bread contain emulsifiers, others do not). The use of the app enabled participants to scan food product label barcodes, meaning foods could be identified as suitable or unsuitable without requiring the knowledge and skills to label-read variations of emulsifier names and E-numbers. Furthermore, this meant participants could be successfully blinded to the diet under investigation, which would enhance a future RCT investigating the clinical effects of a low emulsifier diet.

It is important to evaluate the safety of dietary interventions intended for the treatment of medical conditions. Whilst safety assessments are a key part of medication development [54], exclusion diets are not always stringently evaluated despite potentially impacting negatively upon food-related-QoL, nutrient intake and body weight. A timeframe of 14 days on the low emulsifier diet was selected to ensure that participants had sufficient time to identify major issues with the diet, while minimising participant burden. In fact, we observed that the low emulsifier diet improved Crohn’s disease-related symptoms, patient-perceived disease control and food-related-QoL after two weeks. This finding is promising, but it could reflect a placebo response, which cannot be assessed without a control group [55].

Intakes of energy, carbohydrate, saturated fat, sodium, calcium, vitamin B3, vitamin B12 and alcohol were significantly reduced following the low emulsifier diet. This is unsurprising given that food exclusion diets in IBD can impair nutrient intakes [56], plus habitual intakes of alcohol and energy are lower in IBD compared to healthy populations [57]. Participants reported avoiding ready meals and convenience foods, which could explain the reduction in energy, sodium and saturated fat [58]. These changes in nutrient intakes could be considered a positive health alteration and potentially explain the reduction in gastrointestinal symptoms observed [57]; however, nutrient changes were not clinically significant. The greatest contributor to baseline emulsifier intake were “cereal and cereal grain products”, which likely explains the reduction in carbohydrate intake during the low emulsifier diet. The Specific Carbohydrate Diet (SCD) is purported to improve symptoms by restricting certain dietary carbohydrates [59] and requires avoidance of all cereal grains. It is possible that the observed symptomatic improvements from the limited quality studies of the SCD thus far [60] may relate to a lowering of emulsifier intake, occurring as a consequence of eliminating cereal grain products. If emulsifiers are implicated as a dietary trigger, the SCD may be excessively restrictive as not all grain products contain food-additive emulsifiers, and therefore, it may unnecessarily place patients at risk of nutritional deficiency. Indeed, there are highly promising recent results from studies of other Crohn’s disease diets, such as the Crohn’s Disease Exclusion Diet (CDED) [51] and CD-TREAT diets [52], both of which exclude multiple dietary components including emulsifiers. A common theme among these diets is the reduction in processed foods that contribute substantially to emulsifier intakes, but they also extensively restrict other non-emulsifier containing foods.

One limitation of the current study was that all participants had stable Crohn’s disease, and the feasibility of the diet may be different in people with active Crohn’s disease who may be acutely unwell. Active disease is associated with reduced appetite and other factors that put this group at higher risk of malnutrition than those with quiescent disease [61]. Therefore, despite the high observed acceptability and feasibility of the low emulsifier diet, caution should be applied in generalising these findings to all patients with Crohn’s disease, especially those with additional nutrition needs such as malnutrition and existing dietary restrictions.

A further limitation of this study is that only frequency of consumption of dietary emulsifiers could be assessed and not total quantity consumed. Currently, only the presence of emulsifiers, and not the quantity, is reported on food labels, and therefore, it has not been possible to estimate actual emulsifier intakes. Meals eaten out in restaurants could not be fully assessed for emulsifier content as the ingredients were unknown, and therefore, we have likely underestimated habitual emulsifier intake based on these methods. Nevertheless, even this conservative estimate demonstrates that emulsifier consumption is widespread and frequent in this population, with participants eating an average of two emulsifier-containing meals in a day. Additionally, our analysis demonstrates frequent co-occurrence of emulsifier exposure in the diets of people with Crohn’s disease. In the feasibility study, commonly co-occurring emulsifier consumption could be explained by coexistence in a single food (e.g., bread) or frequently paired foods (e.g., bread with margarine), or it represents a habitual dietary pattern (e.g., patients eating numerous different bakery products). The most commonly co-occurring emulsifiers were from different emulsifier classes. These emulsifiers may therefore have combinatorial effects that are not modelled in current single additive exclusion/addition studies. Due to the novelty of this area of research, there is minimal data on likely human exposure to emulsifiers in the food supply. Therefore, it is challenging to confirm whether emulsifier doses used in the murine study could be achieved in the human population. However, known intakes for some emulsifiers, such as carrageenan, do exceed the safe Acceptable Daily Intake in some individuals.

Studies have shown that people with Crohn’s disease have a keen interest in using diet to treat their disease, with as many as 70% believing that dietary intake is directly responsible for their disease flares [62]. The medical management of Crohn’s disease is not without risk, and patient concerns around side effects commonly prevent or discourage the initiation or adherence to medications [63]. Many patients therefore seek alternative and holistic approaches to manage their condition [64]. We have shown the potential for a range of emulsifier classes to increase gastrointestinal inflammation and that dietary restriction of emulsifiers encompassing these classes is feasible and acceptable in people with Crohn’s disease. A low emulsifier diet could therefore be a low cost and safe therapy in Crohn’s disease.

Evidence implicating emulsifiers in Crohn’s disease is limited to cell line studies, animal models, in vitro human microbiota systems and epidemiological studies. The impact of emulsifiers on disease severity, intestinal inflammation, gut microbiota and barrier function in people with Crohn’s disease is not known. Calls for urgent research on emulsifiers and IBD have been made. The European Food Safety Authority’s Emerging Risks report identified ‘‘food emulsifiers, the gut microbiome and long-term health effects’’ as an emerging risk [65]. The European Crohn’s and Colitis Organisation recently published a report on research gaps in diet and nutrition in IBD, which identified the need for research on emulsifiers in IBD to be undertaken [25]. The present study has not only reinforced the inflammatory potential of several commonly consumed emulsifiers but for the first time successfully designed and tested the feasibility of a low emulsifier diet that can be used in clinical RCTs. We have shown a low emulsifier diet is feasible in people with Crohn’s disease. A human RCT to investigate the clinical effectiveness and physiological effects of a low emulsifier diet in active Crohn’s disease is warranted. RCTs are the gold standard for establishing treatment efficacy, and this design is particularly important for evaluating dietary interventions where participants may be particularly prone to the placebo effect of a novel dietary intervention [55]. Such studies should include an analysis of as the impact on gut microbiota and immune cell signalling that would elucidate the mechanisms underlying the effectiveness of the low emulsifier diet, should it exist.

## 5. Conclusions

The murine study has confirmed the inflammatory potential of food-additive emulsifiers from a range of classes. The feasibility study demonstrated for the first time that a low emulsifier diet is acceptable and safe in people with Crohn’s disease. Further research should now ascertain the therapeutic potential of a low emulsifier diet in Crohn’s disease.

## Figures and Tables

**Figure 1 nutrients-12-02827-f001:**
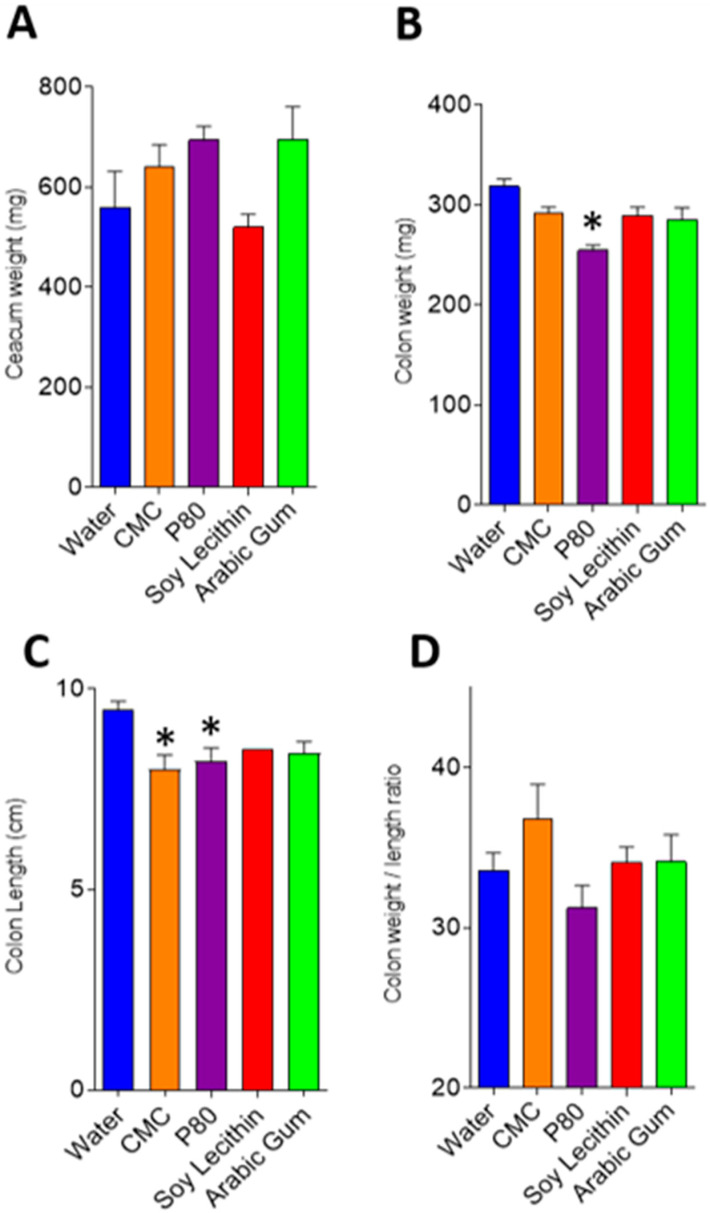
(**A**–**D**): Intestinal weight and length in emulsifier and water-treated mice. Graphs depict mean and standard error for caecal weight (**A**), colon weight (**B**), colon length (**C**) and colon weight/length ratio (**D**). * *p* < 0.05 compared with water (control) following ANOVA and Bonferroni post hoc correction. CMC, carboxymethylcellulose; P80, polysorbate-80.

**Figure 2 nutrients-12-02827-f002:**
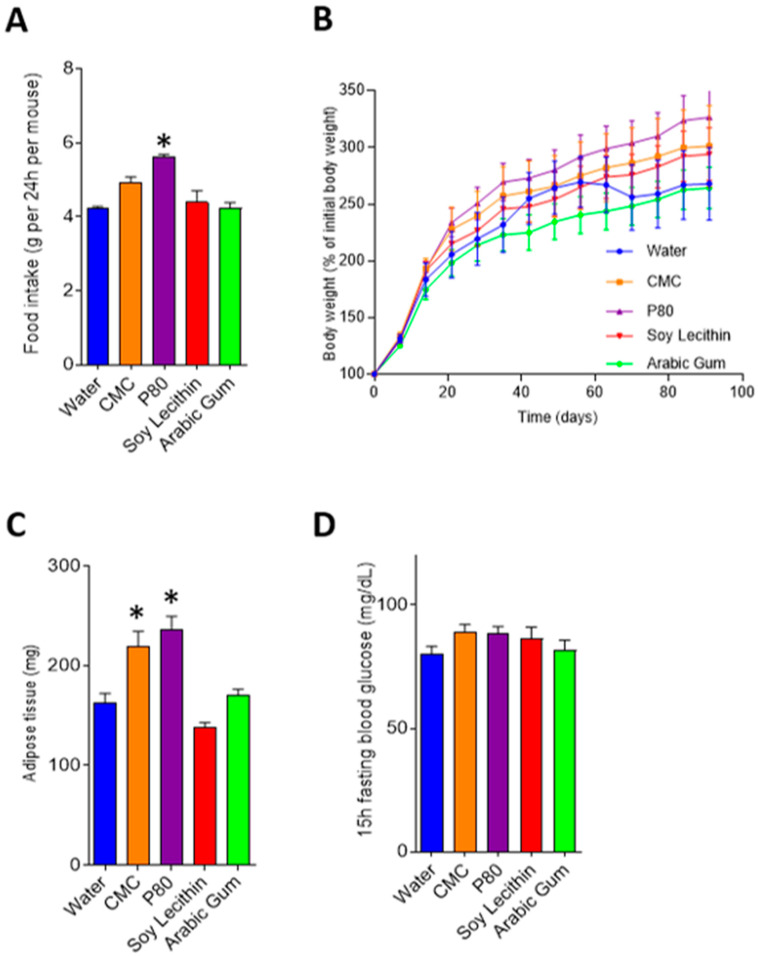
Food intake and metabolic markers in emulsifier and water-treated mice. Graphs depict mean and standard error for food intake (**A**), body weight (**B**), adipose tissue (**C**) and fasting blood glucose (**D**). * *p* < 0.05 compared with water (control) following ANOVA and Bonferroni post hoc correction. CMC, carboxymethylcellulose; P80, polysorbate-80.

**Figure 3 nutrients-12-02827-f003:**
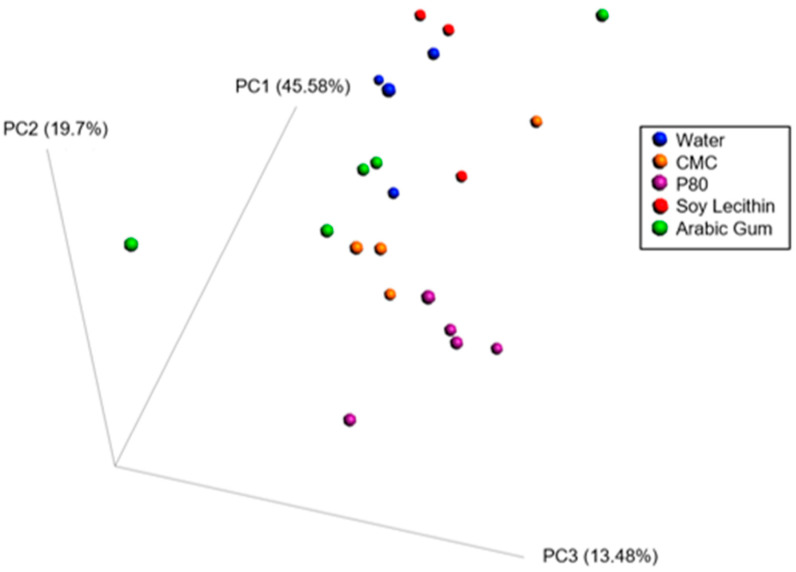
Principle coordinate analysis of the Bray–Curtis distance using a matrix containing morphometric parameters (final body weight percentage, adipose weight, colon weight, colon length, caecum weight, fasting blood glucose level, food intake).

**Figure 4 nutrients-12-02827-f004:**
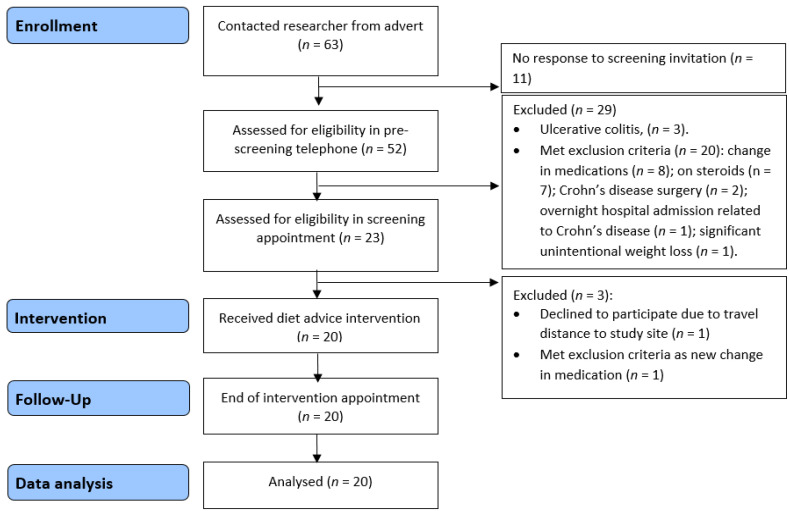
Feasibility study recruitment flow diagram.

**Figure 5 nutrients-12-02827-f005:**
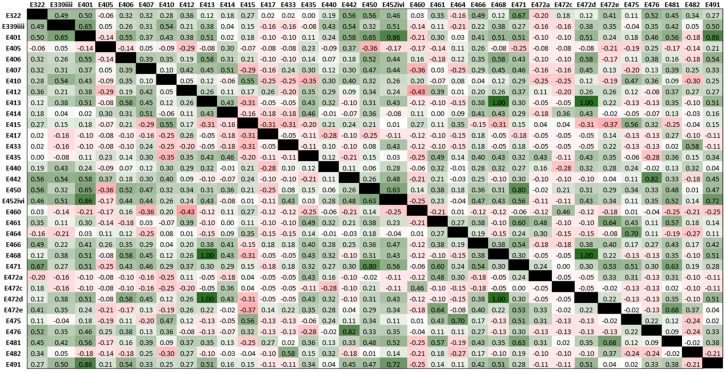
Heatmap showing co-occurrence of dietary emulsifier exposure of participants in the feasibility study. Values are Spearman’s rho correlation coefficients and cell shading indicate the magnitude of correlation co-occurrence between emulsifiers (green is positive correlation; red is negative correlation).

**Table 1 nutrients-12-02827-t001:** Baseline characteristics of 20 participants with Crohn’s disease in the feasibility study.

Characteristic	*N* = 20
Sex, n (%)	
Male	6 (30)
Female	14 (70)
Age (year), mean (SD), min–max	34.9 (10.1), 23–62
Smoking status, n (%)	
Current smoker	0 (0)
Previous smoker	3 (15)
Non-smoker	17 (85)
Disease duration (year), mean (SD), min–max	13.1 (10.8), 2–40
Current medications, n (%)	
5-aminosalicyclic acid	3 (15)
Thiopurines	10 (50)
Biologics	8 (40)

SD, standard deviation.

**Table 2 nutrients-12-02827-t002:** Food groups contributing to frequency of emulsifier exposure in 20 patients with Crohn’s disease in baseline habitual diet.

Food Group ^1^	Contribution to Baseline Emulsifier Intake, N/N (%)	Sub-Food Group Contribution, %
Cereal and cereal products	177/422 (42%)	Bread (21%)
		Biscuits, buns, cakes, pastries and fruit pies (17%)
		Pasta, rice, breakfast cereals and other cereals (4%)
Sugars, preserves and confectionery	83/422 (20%)	Chocolate confectionary (13%)
		Preserves and sweet spreads (6%)
		Sugar confectionary (1%)
Miscellaneous	53/422 (13%)	Savoury sauces (6%)
		Nutrition powders, drinks and protein bar supplements (6%)
		Meat alternatives (1%)
Milk and milk products	42/422 (10%)	Cheese (3%)
		Ice-cream (3%)
		Other/alternative milks (2%)
		Yoghurt (1%)
Fat spreads	26/422 (6%)	Low fat spreads, margarines and oil (6%)
Non-alcoholic beverages	21/422 (5%)	Non-alcoholic beverages (5%)
Savoury snacks	9/422 (2%)	Crisps and savoury snacks (2%)
Meat and meat products	4/422 (1%)	Processed meat (1%)
Fish and fish dishes	4/422 (1%)	Fish and fish dishes (1%)
Vegetables and potatoes	3/422 (1%)	Processed potato products (1%)

^1^ All emulsifier-containing foods consumed by participants in the baseline week (*n* = 422 foods) were categorised into food groups classified according to the National Diet and Nutrition Survey (2011) [49].

**Table 3 nutrients-12-02827-t003:** Nutrient intakes at baseline and following the low emulsifier diet in 20 people with stable Crohn’s disease.

Nutrient	Absolute Intakes, Median (IQR)			Meeting National Guidelines, N (%)		
	Baseline	Low Emulsifier Diet	*p*-Value	Baseline	Low Emulsifier Diet	*p*-Value
Energy (kcal/day)	2055 (587)	1855 (620)	0.025	-	-	
Carbohydrate ^1^ (g/day)	187.0 (66.9)	182.5 (61.8)	0.048	0 (0)	1 (5)	1.000
Fibre ^2^ (g/day)	21.2 (11.1)	21.9 (12.1)	0.627	4 (20)	5 (25)	1.000
Total sugar (g/day)	63.6 (48.9)	65.7 (31.2)	0.126	-	-	-
Free sugars ^1^ (g/day)	27.0 (15.0)	24.8 (25.2)	0.351	7 (35)	8 (40)	1.000
Protein ^3^ (g/day)	81.5 (35.1)	75.3 (47.9)	0.079	19 (95)	18 (90)	1.000
Fat ^1^ (g/day)	89.6 (21.1)	79.6 (45.3)	0.067	5 (25)	3 (15)	0.688
Saturated fat ^1^ (g/day)	31.2 (12.7)	25.0 (14.8)	0.048	5 (25)	5 (25)	1.000
Alcohol ^4^ (g/day)	7.8 (19.2)	0.7 (10.7)	0.016	14 (70)	17 (85)	0.375
Calcium ^2^ (mg/day)	813 (483)	744 (178)	0.015	13 (65)	12 (60)	1.000
Iron ^2^ (mg/day)	12.7 (5.7)	13.5 (4.9)	0.550	11 (55)	9 (45)	0.625
Zinc ^2^ (mg/day)	10.1 (4.3)	9.3 (4.7)	0.391	16 (80)	15 (75)	1.000
Sodium ^5^ (mg/day)	2465 (1276)	1991 (1295)	0.028	9 (45)	12 (60)	0.508
Potassium ^2^ (mg/day)	2813 (1392)	3287 (1141)	0.370	7 (35)	8 (40)	1.000
Chloride (mg/day)	3509 (1541)	3288 (1711)	0.052	-	-	-
Phosphorus ^2^ (mg/day)	1306 (465)	1247 (547)	0.313	20 (100)	20 (100)	N/A
Magnesium ^2^ (mg/day)	320 (128)	356 (166)	0.823	15 (75)	14 (70)	1.000
Iodine ^2^ (µg/day)	102 (70)	81 (64)	0.313	3 (15)	2 (10)	1.000
Selenium ^2^ (µg/day)	55.4 (37.7)	55.0 (41.9)	0.313	8 (40)	6 (30)	0.625
Copper ^2^ (mg/day)	1.29 (1.32)	1.35 (1.05)	0.502	12 (60)	14 (70)	0.688
Manganese (mg/day)	3.8 (1.9)	3.9 (1.7)	0.179	-	-	-
Vitamin A ^2^ (µg/day)	794 (611)	948 (687)	0.391	14 (70)	16 (80)	0.688
Vitamin B1 ^2^ (Thiamine) (mg/day)	1.50 (0.78)	1.46 (0.70)	0.433	20 (100)	19 (95)	1.000
Vitamin B2 ^2^ (Riboflavin) (mg/day)	1.49 (0.71)	1.45 (0.82)	0.627	17 (85)	16 (80)	1.000
Vitamin B3 ^2^ (Niacin) (mg/day)	36.7 (14.8)	33.0 (19.6)	0.021	20 (100)	20 (100)	N/A
Vitamin B5 (Pantothenic acid) (mg/day)	5.51 (3.47)	5.24 (3.16)	0.794	-	-	-
Vitamin B6 ^2^ (Pyridoxine) (mg/day)	1.55 (0.94)	1.73 (0.60)	0.911	18 (90)	17 (85)	1.000
Vitamin B7 (Biotin) (µg/day)	41.8 (23.7)	34.6 (23.0)	0.167	-	-	-
Vitamin B9 ^2^ (Folate) (µg/day)	241.3 (123.3)	247.3 (117.6)	0.627	15 (75)	14 (70)	1.000
Vitamin B12 ^2^ (Cobalamin) (µg/day)	4.98 (3.75)	4.09 (2.86)	0.021	20 (100)	20 (100)	N/A
Vitamin C ^2^ (mg/day)	74.6 (91.2)	87.2 (80.9)	0.433	15 (75)	17 (85)	0.625
Vitamin D ^2^ (µg/day)	3.3 (2.5)	3.2 (4.1)	0.737	2 (10)	0 (0)	0.500

^1^ Public Health England (PHE) (2016) recommendations [38]. ^2^ Reference Nutrient Intakes for men and women from Public Health England (2016) [39]. ^3^ Protein intake values ≥14.5 and <15.5% of food energy are categorised as being within the ~15% guideline, as per Public Health England (2016) [39]. ^4^ UK Chief Medical Officers’ Low Risk Drinking Guidelines (2016) [40]. ^5^ Scientific Advisory Committee on Nutrition [42]. IQR: interquartile range.

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
