# Peer review of "Emulsifiers Impact Colonic Length in Mice and Emulsifier Restriction is Feasible in People with Crohn’s Disease"

_nutrients, 2020, doi:10.3390/nu12092827_

Round 1
Reviewer 1 Report
This manuscript is well written and presents data relating to food additive / dietary emulsifiers and their effect in a murine model and also correlates the effects of low dietary emulsifier intake in a cohort with stable crohn's disease.
In the animal study Colon length was used as a measure of inflammation and all emulsifiers showed an effect on colon length. In the human feasibility trial, a reduction in emulsifier intake was related to reduced crohn's disease symptoms.
Whilst the study is preliminary , the data indicates that specific dietary modification in terms of food additive emulsifiers maybe beneficial in reducing symptoms of crohn's disease.
Author Response
Many thanks for reviewing our manuscript and for your comments.
Reviewer 2 Report
L 107: please specify age/weight of animals
L 113: please specify origin/brand of emulsifiers
L 125: can you add reference of previous published paper for the applied method ?
Data from animal study show that only P80 act as pro-inflammatory agents. Is his data in line with published data ? What about the other emulsifiers that appear to not activate “symptoms” of colonic inflammation?
Line 407: considering the limited number of participants the sentence appears too general
In discussion, the reduction of energy, saturated fat or alcohol assumption are potentially related to clinical symptoms of IBD? I suggest to implement discussion referring to published data
Author Response
Author’s Reply to the Review Report (Reviewer 2)
We would like to thank reviewer 2 for their comments on our manuscript “Emulsifiers impact colonic length in mice and emulsifier restriction is feasible in people with Crohn’s disease”. Thank you for the opportunity to revise the manuscript. Each comment has been addressed in a pointwise manner.
Comment 1: Line 107: please specify age/weight of animals.
Response: Many thanks for highlighting this omission. Mice were 4-5 weeks old and initial bodyweight was mean 12 g. This information has now been added to the manuscript (line 110).
Comment 2: Line 113: please specify origin/brand of emulsifiers
Response: Many thanks for this comment. The origin/brand was Sigma-Aldrich and this information has now been added to the manuscript (line 120).
Comment 3: Line 125: can you add reference of previous published paper for the applied method?
Response: Many thanks for this suggestion to strengthen the methods. A previously published paper relating to this method has now been added to the manuscript (line 127).
Comment 4: Data from animal study show that only P80 act as pro-inflammatory agents. Is his data in line with published data? What about the other emulsifiers that appear to not activate “symptoms” of colonic inflammation?
Response: Many thanks for this comment. It is correct that only a limited number of emulsifiers have been demonstrated to definitively impact inflammation and/or colitis in the murine gut, namely carboxymethylcellulose and polysorbate-80 (Chassaing et al, 2015). Indeed our rationale was to explore that among other emulsifiers.
The following is now reported in lines 384 – 390 to situate the findings of the present study with published literature:
‘‘Reduction in colon length, a proxy measure of intestinal inflammation, was observed following treatment with all the tested emulsifiers, although only carboxymethylcellulose and polysorbate-80 reached statistical significance, although the principal coordinate analysis showed gum arabic had highly variable effects. Previous murine studies have demonstrated the colitis-inducing effects of carboxymethylcellulose and polysorbate-80 (13, 24), but this is the first time other commonly consumed emulsifiers, lecithin and gum arabic, have been tested for their gastrointestinal impact in a murine model.’’
Comment 5: Line 407: considering the limited number of participants the sentence appears too general.
Response: Many thanks for this comment and we wholly agree with your observation. The sentence has now been changed from ‘‘people with Crohn’s disease’’ to ‘‘in this small sample of people with Crohn’s disease’’ (line 409 – 410) to reflect this comment.
Comment 6: In discussion, the reduction of energy, saturated fat or alcohol assumption are potentially related to clinical symptoms of IBD? I suggest to implement discussion referring to published data.
Response: Many thanks for this comment. We agree that energy and alcohol intakes are known to be lower in IBD and have now referenced this fact in the paper (lines 431-432). The point that the reduction in energy, saturated fat and alcohol may be responsible for the reduction in gastrointestinal symptoms is also valid; although in the present study the reduction in these nutrients is not clinically significant (with very small effect sizes) and so no conclusions can be drawn. This point is now discussed in the manuscript too (lines 434-436), many thanks for the suggestion.